# Dynadiff: Single-stage Decoding of Images from Continuously Evolving fMRI

## Abstract

Brain-to-image decoding has been recently propelled by the progress in generative AI models and the availability of large ultra-high field functional Magnetic Resonance Imaging (fMRI). However, current approaches depend on complicated multi-stage pipelines and preprocessing steps that typically collapse the temporal dimension of brain recordings, thereby limiting time-resolved brain decoders. Here, we introduce Dynadiff (Dynamic Neural Activity Diffusion for Image Reconstruction), a new single-stage diffusion model designed for reconstructing images from dynamically evolving fMRI recordings. Our approach offers three main contributions. First, Dynadiff simplifies training as compared to existing approaches. Second, our model outperforms state-of-the-art models on time-resolved fMRI signals, especially on high-level semantic image reconstruction metrics, while remaining competitive on preprocessed fMRI data that collapse time. Third, this approach allows a precise characterization of the evolution of image representations in brain activity. Overall, this work lays the foundation for time-resolved brain-to-image decoding.

## 1 Introduction

**Reconstructing images from fMRI.** The reconstruction of visual perception from brain activity, first started in the early 2000s (Haxby et al., 2001; Carlson et al., 2003; Kamitani & Tong, 2005; Miyawaki et al., 2008), has substantially improved within the past two years (Takagi & Nishimoto, 2023; Ozcelik & VanRullen, 2023; Scotti et al., 2024; Wang et al., 2024b; Benchetrit et al., 2024; Le et al., 2025; Chen et al., 2023b). This recent progress stems from two key factors: the availability of large-scale neuroimaging data in response to natural images (Allen et al., 2022; Hebart et al., 2019), and the emergence of powerful image-generation models (Rombach et al., 2022; Xu et al., 2023; Podell et al., 2023). One dataset has catalyzed brain-to-image decoding: the Natural Scenes Dataset (NSD) (Allen et al., 2022) is the largest dataset of brain responses to natural images, and consists of subjects watching 70K images over 40 sessions, while their brain activity is recorded with ultra-high field (7T) functional Magnetic Resonance Imaging (fMRI).

Typically, brain-to-image decoding consists of multiple steps. First, the images successively seen by the participants are embedded into pretrained computer vision models. Second, a deep neural network is trained to transform the brain responses into these image representations. Third, the predicted representations are used to condition a pretrained image-generation model. Finally, several studies generate multiple images, and use an ad-hoc scoring to select the best image (Kneeland et al., 2023; Scotti et al., 2023; Wang et al., 2024b). This approach, first based on principal components analyses (Cowen et al., 2014), auto-encoders (Han et al., 2019) and generative adversarial networks (Seeliger et al., 2018; Shen et al., 2019; Gu et al., 2022; Ozcelik et al., 2022) now primarily relies on diffusion models (Takagi & Nishimoto, 2023; Ozcelik & VanRullen, 2023; Scotti et al., 2024; Wang et al., 2024b; Benchetrit et al., 2024; Le et al., 2025).

**Challenge 1: time-collapsing fMRI preprocessing.** The best brain-to-image reconstructions to date have been achieved on the Natural Scenes Dataset by decoding *time-collapsed* fMRI 'beta values'. These are derived by fitting a generalized linear model (GLM) across time to isolate the brain's response to each image, and entail multiple challenges. First, the type of beta values used in prior state-of-the-art studies completely discards the time dimension of fMRI data. Second, most

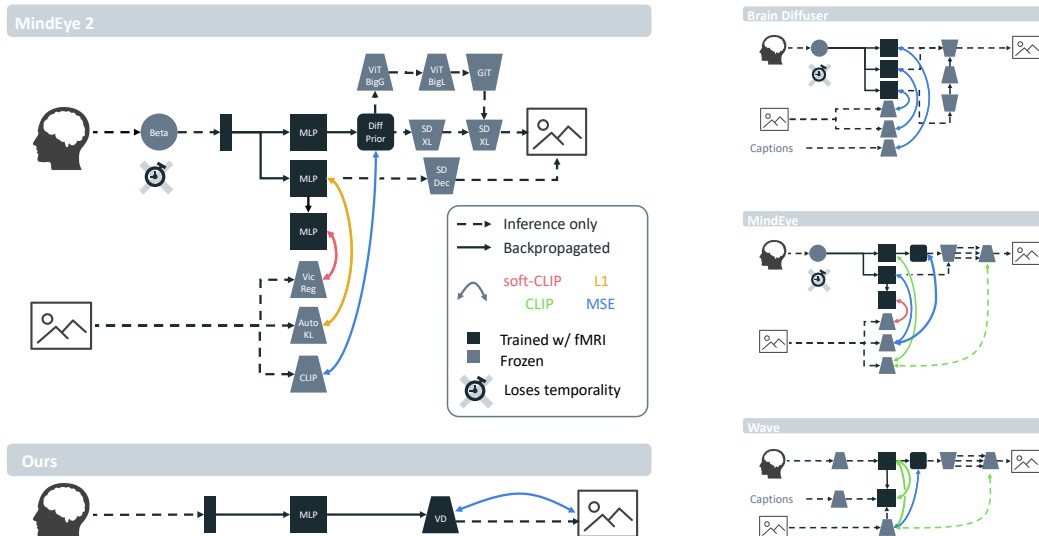

Figure 1: Schematic bird's-eye view of four seminal fMRI-to-image architectures: Brain-Diffusers (Ozcelik & VanRullen, 2023), MindEye1 (Scotti et al., 2023), WAVE (Wang et al., 2024b), MindEye2 (Scotti et al., 2024). They all consist of multiple independent training modules, and can't be trained in a single stage. Except WAVE, they use a preprocessing of fMRI data which collapses time. We illustrate the simplicity and time-resolved capability of our approach, Dynadiff, trained in a single stage on timeseries of fMRI activity, in comparison to these pipelines.

studies average these beta values across multiple presentations of the same image. As a result, this preprocessing of fMRI data severely restricts the ability to perform time-resolved image decoding.

**Challenge 2: complex multi-stage decoding pipelines.** The complexity of decoding pipelines has substantially increased in the past two years. State-of-the-art models now consist of up to four independent stages including pretrained fMRI encoders (Chen et al., 2023b; Wang et al., 2024b; Huo et al., 2024), contrastive learning (Chen et al., 2023b; Wang et al., 2024b; Scotti et al., 2024; Xia et al., 2024a), diffusion priors (Scotti et al., 2024; 2023; Wang et al., 2024b), automatic image captioning (Ferrante et al., 2023; Scotti et al., 2024), control nets (Ferrante et al., 2023; Huo et al., 2024) and post-candidate selection (Kneeland et al., 2023; Scotti et al., 2023). Many of these steps optimize, either separately or jointly, a variety of losses, supported by advanced data augmentation techniques (Scotti et al., 2024; Wang et al., 2024b). Figure 1 illustrates four seminal pipelines to highlight both the high complexity of modern brain-to-image decoders, and the increase of this complexity over the years (Ozcelik & VanRullen, 2023; Scotti et al., 2023; 2024; Wang et al., 2024b). For example, the state-of-the-art pipeline MindEye2 (Scotti et al., 2024) requires pretraining a custom image generation model (SDXL-UnCLIP), captioning its outputs and refining reconstructions using SDXL. Even the lower-performing but arguably simple Brain-Diffuser (Ozcelik & VanRullen, 2023), which only uses ridge regression models, also requires two independent training / inference stages, for low- and high-level image reconstruction respectively. Overall, the simplification of the many manual feature engineering steps, often promised by deep learning approaches, seems here to fall short.

**Our main contributions.** We introduce Dynadiff, a pipeline to reconstruct images from dynamically evolving fMRI signals. First, it uses only a single-stage of training and inference, contrasting with the complexity of previous approaches. Second, it outperforms state-of-the-art models on fMRI time-series from the Natural Scenes Dataset. Third, the dynamic nature of our approach enables an accurate description of how image representations change in brain activity over time.

## 2 METHOD

### 2.1 PROBLEM STATEMENT

Our goal is to reconstruct images from continuously evolving BOLD fMRI signals recorded while participants watched natural images. Let $W(s, t, d)$ denote the time-window of $d$ seconds starting $t$

seconds after the onset of an image stimulus $s$. Since fMRI volumes are acquired at frequency $f = \frac{1}{\text{TR}}$ (see Section 2.3), this time window corresponds to a time-series $X \in \mathbb{R}^{C \times T}$ of $T \approx f \cdot d$ volumes of $C$ fMRI voxels each (where $C$ typically varies across participants due to anatomical differences). Given fixed $t$ and $d$, we aim to reconstruct $s$ given $X$. To tackle this task, we propose Dynadiff, which directly fine-tunes a pretrained image-generation diffusion model with the fMRI signals, as illustrated in Figure 1. Specifically, we design a brain module that projects $X$ to a conditioning embedding of the diffusion model. This brain module is jointly trained with the diffusion model to learn to reconstruct realistic and consistent images. We give more details about this brain module, explain how we adapt the pretrained diffusion model and the joint training in Section 2.2.

## 2.2 DYNADIFF

**Brain Module.** The brain module projects fMRI data $X$ to the conditioning space of the image-generation model and is shown in Figure 2. It consists of a subject-specific linear layer $\mathbf{S} : \mathbb{R}^C \to \mathbb{R}^{1552}$ (Défossez et al., 2022) that projects each fMRI volume (of $C$ voxels) to 1,552 channels, while keeping the same number $T$ of fMRI time samples. This layer outputs a vector $Z \in \mathbb{R}^{1552 \times T}$.

Then, $Z$ is passed to a timestep-specific linear layer, that applies a distinct set of weights to each time sample. This is followed by a layer normalization, a GELU activation and dropout ($p = 0.5$). Next, a linear temporal aggregation layer merges the temporal dimension. Finally, an additional linear layer outputs fMRI embeddings with the same shape as the conditional embedding of the image-generation model: 257 patches and 768 channels. Our brain module has around 400M parameters.

**Brain-conditioned diffusion model.** For simplicity, we use the same pretrained latent diffusion model as in Ozcelik & VanRullen (2023); Scotti et al. (2023). This conditional image-generation model is based on a U-Net architecture (Ronneberger et al., 2015) and was trained to synthesize images conditioned on texts and images. These two prompts are first projected to token embeddings using the text and image encoders from CLIP (Radford et al., 2021). Then, these embeddings are processed through cross-attention layers, which are present at different feature-map scales of the U-Net. To condition the diffusion model on fMRI data, we replace image embeddings with the output from the brain module, and provide null text embeddings[*]. This approach enables us to leverage the pretrained generative model's ability to synthesize high-quality images.

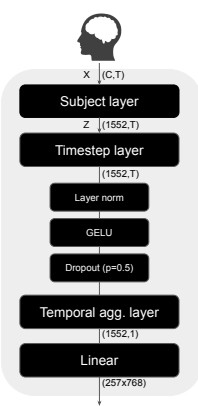

Figure 2: The architecture of our brain module, corresponding to the only MLP block of our approach (Figure 1).

**Single-stage training.** We jointly train the brain module and the brain-conditioned diffusion model to reconstruct seen images from fMRI data. The brain module and LoRA adapters (Hu et al., 2021) for the diffusion model's cross-attention layers are trained from scratch while the generation model weights are left untouched ($\sim 25$M parameters). We use the standard diffusion loss to optimize the model weights. Additionally, we use (1) bicubic sampling (Mou et al., 2024), which involves more frequent sampling of early timesteps during training and (2) an offset noise[†]. Finally, to enable classifier-free guidance at inference time, we remove the brain-conditioning in 10% of training iterations, and replace it with a constant learned embedding instead. Further details on training optimization can be found in Appendix B.

**Inference.** We reconstruct an image $I$ from a time-series $X$ of fMRI volumes as follows. First, we apply the brain module on $X$ to obtain fMRI embeddings $Z$. Then, we sample an initial random gaussian noise $\epsilon$, and provide both $Z$ and $\epsilon$ to the diffusion model's U-Net to start the denoising process; we use a DDIM scheduler with 20 denoising steps and a classifier-free-guidance scale of 3. This process yields a denoised latent embedding, which is then passed to the diffusion model's

---

[*]We empirically observed that replacing text embeddings didn't boost performance.

[†]https://www.crosslabs.org/blog/diffusion-with-offset-noise

autoencoder to produce a reconstruction of $I$. To ensure reproducibility, we will make our model's inference code publicly available upon publication.

## 2.3 EXPERIMENTAL SETTING

**Dataset.** We here use the Natural Scenes Dataset (Allen et al., 2022). Eight healthy volunteers participated in this dataset, six females and two males, with a mean age range of 19 to 32 years. Each volunteer underwent 30 to 40 fMRI sessions, each lasting approximately one hour. Consistent with prior research (Ozcelik & VanRullen, 2023; Scotti et al., 2023; 2024), we focus on the data from subjects who completed all 40 recording sessions, *i.e.*, subjects 1, 2, 5, and 7. Each participant viewed 10,000 unique images from the MS-COCO dataset (Lin et al., 2014), and each unique image was presented three times over the 40 sessions. Of these images, 9,000 are used for training, while a shared set of 1,000 images, viewed by all subjects, is reserved for testing. Each image is displayed for 3 s, followed by a 1 s blank interval before the next image presentation. To maintain the time-resolved compatibility of our approach, we don't average the repetitions to the same images, neither during training nor testing. This results in a training dataset of 9000 x 3 = 27,000 trials and a test dataset of 1000 x 3 = 3,000 trials for each subject. We evaluate metrics on the set of 1,000 unique test images, by randomly selecting one test presentation out of three, for each unique test image.

**fMRI preprocessing.** We use the "standard-resolution" timeseries of BOLD fMRI volumes provided by the NSD authors (TR=1.3 s, 1.8 mm isotropic resolution) (Allen et al., 2022). As described in Allen et al. (2022), these data are computed from raw functional timeseries by applying (i) a temporal upsampling, which corrects slice-time differences, and (ii) a spatial resampling, which corrects for head motion, EPI distortion, gradient nonlinearities, and scan session alignment. Compared to the computation of "averaged beta values" used in previous studies (Scotti et al., 2023; 2024; Huo et al., 2024), this preprocessing does not collapse the time domain. It can be reproduced from scripts from the NSD GitHub repository [‡], and deliberately leaves out high-pass filtering, nuisance regression, to avoid unnecessary underlying assumptions. Following previous works (Ozcelik & VanRullen, 2023; Scotti et al., 2023; 2024), we restrict fMRI volumes to the `nsdgeneral` subset (Allen et al., 2022), a Region of Interest manually-outlined on 'fsaverage' located in the posterior cortex (Fischl et al., 1999). Then, we remove low-frequency noise in the fMRI signal using an additional detrending step: we fit a cosine-drift linear model to each voxel in the time series, and subtract it from the raw signal. Finally, each voxel time-series is z-score-normalized. This preprocessing is used for training and evaluating Dynadiff and the other baselines reported for fMRI BOLD time-series. Please refer to Appendix D for an ablation on fMRI preprocessing.

**Reconstructing images over time.** As explained in Section 2.1, our models are trained on fixed fMRI time windows $W(s, t, d)$. To evaluate how well these models generalize across time, we also infer reconstructions from time windows shifted with regard to the image onset. More precisely, at test time, instead of conditioning our model on the usual training time-window, we evaluate it on a shifted window $W(s, t + \delta, d)$ in which $\delta$ can take positive or negative values. Please note that even if the window starts before the image onset (*i.e.*, $t + \delta$ is negative), the fMRI timeseries may still contain relevant information about the image $s$, depending on the duration $d$ of the window.

**A split for time-resolved decoding.** NSD interleaves train and test image presentations. For example, an image from the train set can be presented immediately after an image from the test set. Consequently, evaluating decoding performance over a succession of images requires a re-definition of the train/test splits. For Figures 3 and 5, where we directly report the generation of successive images, we thus create a new *time-resolved* train/test split that ensures that successive trials belong to the same split. For this, we used, for each subject separately, 45 fMRI recording runs for the test set. We then use the 435 remaining runs for training. This split yields approximately 27,000 training trials and 3,000 testing trials (making it aligned with the sizes of the original NSD split). Note that we use this split exclusively for Figures 3 and 5, as it is the only analysis that shows results for successive images. All the other evaluations are conducted with the standard splits defined by NSD.

**Evaluation metrics.** Following previous works *e.g.* Ozcelik & VanRullen (2023); Scotti et al. (2024); Wang et al. (2024b), we assess low-level image similarity using PixCorr (pixel-wise correlation), SSIM (Structural Similarity Index Metric) and Alexnet(2/5), and high-level resemblance include CLIP, Inception, Efficient-Net, and SwAV. We complement our analysis with two additional metrics,

---

[‡] https://github.com/cvnlab/nsddatapaper

which are computed trial-wise, *i.e.*, : for each unique test image $I$, we evaluate the metric on the pair consisting of $I$ and its reconstruction from a randomly sampled repetition of $I$. Then, the results are averaged over the 1,000 unique test images. First, we use the DreamSim metric (Fu et al., 2023), which leverages a mixture of pretrained backbones trained on a human similarity-judgments dataset. Second, we use mIoU over semantic-segmentation masks for semantic consistency and interpretability. It is computed by passing each image and its reconstruction through a semantic segmentation network and comparing their predicted semantic maps. We use ViT-Adapter (Chen et al., 2023a) as segmentation model. All metrics are computed after resizing images to 224×224 pixels.

**Baselines.** We compare our method to the seminal work of Brain-Diffuser (Ozcelik & VanRullen, 2023) as well as three state-of-the-art approaches: (i) MindEye (Scotti et al., 2023) and MindEye 2 (Scotti et al., 2024), originally designed for time-collapsed fMRI 'beta-values', and (ii) WAVE, which uses timeseries of BOLD fMRI signal as input. When applicable, we provide the details of how we adapted these methods to time series of NSD in Appendix A.

## 3    RESULTS

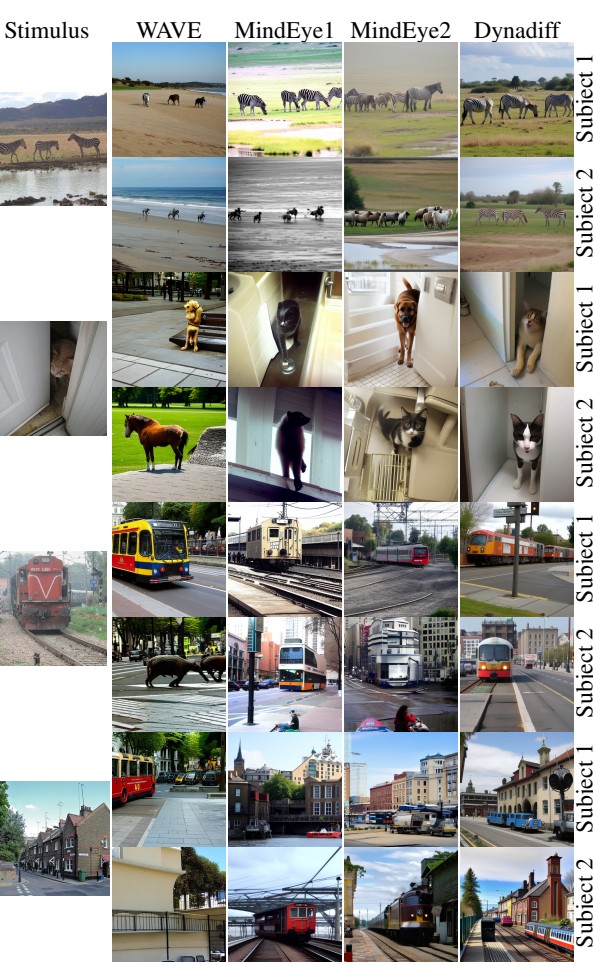

Figure 3: Qualitative comparisons of Wave, MindEye1, MindEye2 and our model on the NSD dataset. We display the image stimuli on the left column and the next columns show WAVE, MindEye1, MindEye2 and our model successively.

Table 1: Comparison to baselines on the Natural Scenes Dataset. We average results across subjects 1, 2, 5 and 7 (as done in the corresponding studies). Notably we evaluate all the methods in single trial, without averaging same-image repetitions and provide SEM.

| Baselines | Low-level | | | | Semantic and High-level | | | | | |
|---|---|---|---|---|---|---|---|---|---|---|
| | ↑SSIM | ↑PixCorr | ↑AlexNet(2) | ↑AlexNet(5) | ↑CLIP-12 | ↑Incep | ↓Eff | ↓SwAV | ↑mIoU | ↓DreamSim |
| Wave | 0.15 ± 0. | 0.07 ± 0. | 68.99 ± 0.71 | 77.44 ± 0.81 | 76.76 ± 0.41 | 73.24 ± 0.67 | 0.85 ± 0. | 0.53 ± 0. | 2.24 ± 0.14 | 68.67 ± 0.33 |
| Brain-Diffusers | 0.21 ± 0. | 0.21 ± 0.01 | 89.41 ± 1.16 | 92.68 ± 0.88 | 85.36 ± 1.05 | 84.06 ± 1.07 | 0.80 ± 0.01 | 0.48 ± 0.01 | 7.73 ± 0.28 | 60.39 ± 0.86 |
| MindEye1 | 0.31 ± 0.01 | **0.27 ± 0.01** | 91.45 ± 2.21 | 95.45 ± 0.62 | 91.11 ± 0.61 | 88.78 ± 0.92 | 0.73 ± 0.00 | 0.40 ± 0.00 | 7.55 ± 0.35 | 57.68 ± 0.67 |
| MindEye2 | **0.36 ± 0.00** | 0.24 ± 0.01 | 94.15 ± 0.99 | 97.34 ± 0.50 | 90.38 ± 0.80 | 89.47 ± 0.89 | 0.71 ± 0.01 | 0.38 ± 0.01 | 8.15 ± 0.45 | 56.28 ± 0.95 |
| Dynadiff | 0.34 ± 0.00 | 0.21 ± 0.01 | **95.82 ± 0.82** | **98.20 ± 0.41** | **93.53 ± 0.67** | **91.30 ± 0.74** | **0.68 ± 0.01** | **0.36 ± 0.01** | **8.50 ± 0.41** | **52.52 ± 0.97** |

## 3.1 DIFFUSION-BASED IMAGE RECONSTRUCTION

**Comparison to Baselines.** Table 1 reports quantitative metrics using the four subjects of NSD, for our model and competing approaches, evaluated in single-trial (using individual and non-pooled fMRI timeseries). We also compute Standard Error of the Mean (SEM) computed across the four subjects. Brain-Diffuser, MindEye, MindEye2 and our model were trained with a time-window of $6 \cdot TR$ (8 s), starting 3 seconds after image onset. To follow the original approach of Wang et al. (2024b), WAVE was trained with windows starting one TR (1.3 s) after image onset and lasting $5 \cdot TR$ (6.5 s). Our approach outperforms other methods including the state-of-the-art MindEye2. Indeed, our model improves by respectively 1.67 and 0.86 points on Alexnet(2) and AlexNet(5) compared to MindEye2, showing that it better preserves low-level contents (*e.g.* color, texture). We also achieve a 3.76-point improvement over MindEye2 on DreamSim and a 3.25-point increase on CLIP-12. This emphasizes our model's ability to correctly decode object semantics and positions. Additionally, the performance for each subject is reported in Table 6. To complete our analysis, we report the evaluation of these models on i) fMRI test-trial windows averaged across same-image repetitions (in Table 7), and on ii) the "average beta-values" commonly used in previous studies on NSD (in Table 4). The latter is obtained by treating a beta-value as a fMRI time-series with a single timestep. Please refer to our appendix in Appendix E for our analysis on cross-subject decoding.

**Qualitative comparison.** Figure 3 displays a comparison of reconstructions from Subjects 1 and 2 with our approach and other methods, using trials of four different images. Our approach shows superior alignment between the seen and reconstructed stimuli. For instance, in the first row, the positioning and size of the zebras in the reconstructed image more closely resembles their arrangement in the stimulus. Also, in the second row, our model accurately places a cat at the doorway, demonstrating improved scene compositionality. Additional qualitative results for the four subjects are provided in Figures 7 and 8 of Appendix F.

**Time-resolved decoding of images.** In Figure 4 and Figure 5, we evaluate our model in the time-resolved setting by decoding images through time. We use the new time-resolved train/test split defined in Section 2.3, which ensures that only image stimuli from the test set are input to the decoder at inference. We consider two evaluation settings. In the first setting called "General", we train a model $M_{gen}$ on fMRI time windows $W(s, t, d)$ with fixed $t = 3$ s and $d = 8$ s (~6 TRs). At test time, we evaluate $M_{gen}$ on shifted (test) windows $W(s, t + \delta, d)$, assessing its abilities to generalize to new timesteps. Figure 4 shows seven columns of reconstructed stimuli respectively obtained for $\delta = k \cdot TR$, with $k \in \{-3, -2, \ldots, 3\}$. Additionally, Figure 5 examines 16 different shifted windows corresponding to $k \in \{-6, -5, \ldots, 9\}$. Please note that the x-axis values correspond to the upper end of the shifted time windows (meaning $x = t + \delta + d$). In the second setting named "Specialized", we train for each $\delta$ a model $M_{t+\delta}$ on time windows $W(s, t + \delta, d)$ with fixed $d = 8$ s.

For the "General" model $M_{gen}$, the two extreme values $\delta = -3 \cdot TR$ and $\delta = 3 \cdot TR$ correspond precisely to the time windows $W(\cdot, t, d)$ of the previous and next stimuli presentations respectively, and we observe that the model effectively tends to decode the previous and next images. Furthermore, we notice that $M_{gen}$ generalizes well to time windows unseen at training: it is able to reconstruct quite well stimuli presented at timestep $t$ from the shifted window $W(s, t', d)$ for values of $t'$ that are close enough to $t$. However, the best performance across all timesteps is clearly achieved by using at $t'$ the "Specialized" model $M_{t'}$, which was trained to decode at the stimulus relative onset $t'$. This phenomenon is also illustrated in Figure 5 in which we compute the temporal evolution of SSIM, AlexNet(2), CLIP and mIoU with the specialized models $M_t$ and the general model $M_{gen}$. We observe that we start to decode the image stimulus 3 s after it was shown to the participant, which is

coherent with the Hemodynamic Response Function's (HRF) profile. We can still decode the stimulus reasonably well by taking a time window starting 10 s after presentation using the specialized models.

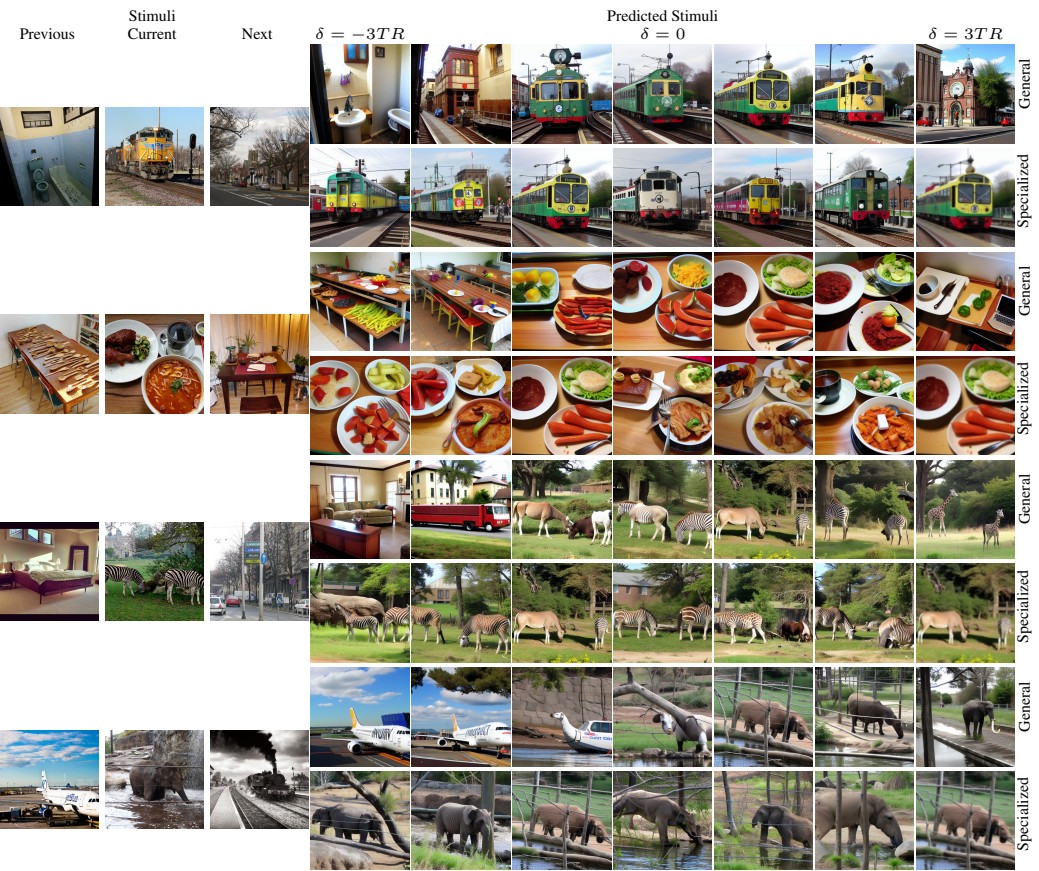

Figure 4: Real-time decoding of images using our specialized or general models Dynadiff. The "General" model $M_{gen}$ is trained on time windows $W(s, t, d)$ (with $t = 3$ s and duration $d = 8$ s) and we evaluate its generalization capabilities by reconstructing images from shifted windows $W(s, t + \delta, d)$. In the "Specialized" setting , we train a separate model for each shift $t + \delta$ on windows $W(s, t + \delta, d)$. This means that each column corresponds to a different model. Since participants see a stimulus every 4 seconds, $\delta = -3 \cdot \text{TR}$ and $\delta = 3 \cdot \text{TR}$ correspond to the windows of the previous and next image presentations respectively. As expected, $M_{gen}$ can decode these images quite well.

## 3.2 ABLATIONS

**Ablation on time window duration.** We perform an ablation study on the duration $d$ of the time window $W(s, t, d)$. Specifically, we fix $t = 3$ s and compare durations $d \in \{1 \cdot \text{TR}, \dots, 6 \cdot \text{TR}\}$. We train one model for each of these six time windows. Then, we conduct a quantitative evaluation by computing low-level metrics (AlexNet2 and AlexNet5) and high-level metrics (CLIP and Inception). Figure 6 shows the evolution of these scores as a function of $d$. We observe that almost optimal performance is obtained with a window duration of $3 \cdot \text{TR}$ (3.9 s) while performance can be slightly enhanced by extending the duration to $6 \cdot \text{TR}$ (7.8 s).

**Ablation on brain module design.** In Table 2, we analyze the effect of specific components of our brain module. First, we investigate the role of the time-specific layers. The first row of Table 2 shows that replacing them with a single linear layer shared across all fMRI time samples reduces performance by 2.95 CLIP-12 points and 1.33 AlexNet(2) points. This implies that these layers are important for allowing the model to leverage the information encoded in the fMRI brain volumes independently. Second, we examine how the position of our time-aggregation layer affects performance. The second row of Table 2 shows that relocating the component from the output layer to the input layer of the brain module decreases performance. This decline might occur because our model is more effective at capturing dynamics of fMRI data when it undergoes additional processing.

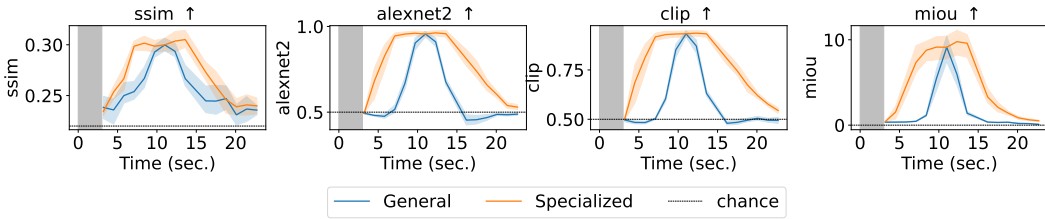

Figure 5: Each point is obtained by reconstructing images from a different fMRI time window $W(s, t + \delta, d)$. We fix $t = 3$ s and duration $d = 8$ s and vary $\delta$ as explained in Section 3. The x-axis represents the end time of time window, *i.e.*, $t + \delta + d$ Orange curve is obtained with specialized models trained specifically for each time window $W(s, t + \delta, d)$ while the blue curve displays the performance results of a general model trained at $W(s, t, d)$ and evaluated at shifted test windows. We provide standard error of the mean on the four NSD subjects. The shaded gray area indicates the 3 sec. interval during which images were presented to the participants.

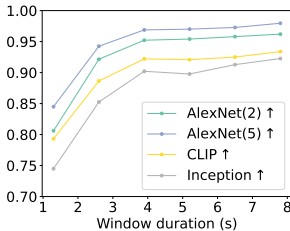

Figure 6: Evolution of AlexNet(2/5), CLIP and Inception metrics when varying the time window duration used to train Dynadiff. More precisely, we use time windows $W(s, t, d)$ that start $t = 3$ s after the stimulus onset and vary their duration $d \in \{1 \cdot \text{TR}, \ldots, 6 \cdot \text{TR}\}$, with TR $= 1.3$ s.

**Ablation on diffusion model finetuning.** In Table 3, we explore which layers of the latent diffusion model are the most useful to finetune, by computing SSIM, AlexNet(2), CLIP-12 and mIoU metrics. First, we consider finetuning all the diffusion model's weights ($\sim$1.1B parameters). This leads to overfitting quickly and poor performance. Second, we experimented with finetuning (i) all the linear layers of the diffusion model (totalizing $\sim$500M parameters), or (ii) all cross-attention linear layers ($\sim$100M). Both options led to suboptimal results. Finally, keeping the entire diffusion model frozen (*i.e.*, training only the brain module with the diffusion loss) also performed worse comparing to adding LoRA adapters to the diffusion model's cross-attention layers.

Table 2: Ablation on brain module design. We report SEM computed across the four subjects.

| Timestep layer | Temporal agg. layer | Low-level | | High-level | |
|---|---|---|---|---|---|
| | | ↑SSIM | ↑AlexNet(2) | ↑CLIP-12 | ↑mIoU |
| ✗ | OUT | 0.29 ± 0.00 | 94.87 ± 0.88 | 90.44 ± 0.51 | 6.95 ± 0.92 |
| ✓ | IN | 0.28 ± 0.00 | 94.31 ± 0.92 | 91.69 ± 0.48 | 7.53 ± 0.86 |
| ✓ | OUT | **0.34 ± 0.00** | **94.67 ± 0.91** | **97.45 ± 0.53** | **8.50 ± 0.97** |

Table 3: Ablation on diffusion model training. We report SEM computed across the four subjects.

| Trainable Layers | ♯Params | Low-level | | High-level | |
|---|---|---|---|---|---|
| | | ↑SSIM | ↑AlexNet(2) | ↑CLIP-12 | ↑mIoU |
| All | 1.1B | 0.29 ± 0.01 | 90.42 ± 0.86 | 88.46 ± 0.52 | 7.56 ± 0.93 |
| Linear | 500M | 0.36 ± 0.00 | 95.74 ± 0.84 | 90.27 ± 0.56 | 8.29 ± 0.98 |
| Cross-Attn | 100M | 0.33 ± 0.00 | 95.63 ± 0.89 | 91.80± 0.48 | 8.08 ± 0.90 |
| ∅ | 0 | 0.32 ± 0.00 | 92.78± 0.92 | 92.77± 0.45 | 7.12 ± 0.99 |
| LoRA on Cross-Attn | 25M | **0.34 ± 0.00** | **94.67 ± 0.91** | **97.45 ± 0.53** | **8.50 ± 0.97** |

## 4 DISCUSSION

**Contributions and related works.** The study offers three major contributions. First, we present a considerably simplified decoding pipeline. Contrasting with recent proposals (Ozcelik & VanRullen,

2023; Scotti et al., 2023; 2024; Chen et al., 2023b; Wang et al., 2024b; Huo et al., 2024), Dynadiff neither depends on (1) a pretrained fMRI encoder (2) an alignment stage between fMRI and pretrained embeddings (3) the post-selection and refining of image generation (4) independent low- and high-level reconstructions. Instead, Dynadiff is trained in a single stage, with a single diffusion loss.

Second, Dynadiff obtains state-of-the-art performance on continuously-evolving fMRI BOLD signals. In particular, the fMRI preprocessing used in this study contrasts with current decoders, especially those trained on the NSD dataset (Ozcelik & VanRullen, 2023; Scotti et al., 2023; 2024), which use 'beta values' obtained from a GLM preprocessing stage that removes the time dimension of fMRI recordings. Notably, even though some previous studies employ fMRI preprocessing which maintain temporal dynamics (Wang et al., 2024b), their reconstruction quality still does not match that achieved with betas. Leveraging the temporal dimension of fMRI recordings naturally suggests progressing from decoding static images to decoding videos. Several studies have now started to develop pipelines to decode videos from 3T fMRI (Chen et al., 2023b; Nishimoto et al., 2011; Fosco et al., 2024). However, these approaches often rely on time-collapsed beta-values, involve multiple training stages and are typically based on a much smaller amount of data. Also, since NSD facilitated the most impressive image reconstructions to date, our approach focuses on temporally-resolved decoding of static images, to establish a robust baseline before advancing to the more complex video objectives. Similarly, the decoding of time-resolved brain activity is now generalized to a variety of tasks such as speech perception (Défossez et al., 2022; Tang et al., 2023; d'Ascoli et al., 2024), continuous visual perception and behavior (Schneider et al., 2023). A major goal for future research will be to unify these efforts into a general architecture for decoding the representation of the brain.

Third, beyond its decoding performance, the present approach enables a temporal analysis of image representations in brain activity. More precisely, it reveals an unexpected phenomenon. The decoder trained at a given time sample with respect to image onsets, can decode the image for a relatively short amount of time. Yet, outside of this generalization window, it is still possible to decode the image, but thanks to a decoder trained specifically around this time point. This result is clearest in Figure 3, where it is possible to decode the current image with specialized decoders, at moments where the generalized decoder reconstruct either the preceding or the next image. This result suggests that the neural patterns that represent images in the fMRI continuously change over time, and allow the simultaneous decoding of successive images. This dynamic coding, typically observed with electrophysiology or M/EEG (King & Dehaene, 2014), may thus apply to fMRI, in spite of its notoriously low temporal resolution. If confirmed, this result would indicate that dynamic coding may be a general process to represent the succession of images, while avoiding their mutual interference.

**Limits.** The present approach remains limited in three ways. First, while NSD is the largest fMRI dataset of individual responses to images (Allen et al., 2022), it was highlighted that the image distribution of images presented to participants tends to follow stereotypical clusters (Shirakawa et al., 2024). It will thus be important to validate the present approach to potentially less biased datasets. Second, Dynadiff is trained on preprocessed fMRI data, a step used to remove movement and cardiac artifacts, align MRI segments, select the relevant voxels. This preprocessing step would likely improve if it was replaced with a foundational model of brain activity, similarly to Wang et al. (2024b); Dadi et al. (2020); Thomas et al. (2022). Third, Dynadiff currently requires many data per participants. It is not adapted to generalize decoding to participants that are not in the training set. Whether it is possible to reliably reconstruct images from any brain remains an open challenge.

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

# A    BASELINES

**MindEye**   The MindEye (Scotti et al., 2023) and MindEye 2 Scotti et al. (2024) approaches reconstruct the stimuli from NSD using volumes of time-collapsed fMRI averaged beta-values restricted to the Region Of Interest 'nsdgeneral'. We keep the exact same stages, hyperparameters and architectures as in Scotti et al. (2023; 2024), with one exception. Since we train and evaluate on BOLD timeseries of NSD (also restricted to 'nsdgeneral'), we have to slightly adapt the architectures to account for the additional time dimension: given an input BOLD fMRI time-series with $T$ time samples of $C$ voxels each, we flatten this window into a vector of size $C \times T$ pass it to the first linear layer of MindEye1 and MindEye2, whose size is increased from $C$ to $C \times T$. The rest of the architecture is left untouched.

**WAVE**   The approach of WAVE Wang et al. (2024b) reconstructs stimuli from timeseries $X \in \mathbb{R}^{1024 \times 5}$ of 5 consecutive TRs of BOLD fMRI volumes, picked at one TR after stimulus onset, and registered to the DiFuMo-1024 atlas Dadi et al. (2020). In a first 'contrastive' stage, fMRI representations are extracted from this atlas using an off-the-shelf fMRI encoder pretrained in a self-supervised fashion Thomas et al. (2022). This encoder is fine-tuned, together with a prompt learning model, with a modality-wise contrastive loss. In a second and independent 'decoding' stage, the same fMRI encoder is tuned again with a diffusion prior module (trained from scratch) to obtain representations that condition an image-generation model Xu et al. (2023). Then, the latter is used to infer two candidate reconstructions for $X$, among which a top one is selected for output (via clip-scoring against the CLIP-aligned fMRI representation).

We evaluate the WAVE method on the NSD dataset as follows: (i) timeseries of BOLD volumes are mapped to the DiFuMo-1024 space using the default parameters of Dadi et al. (2020), and windows of 5 TRs of fMRI starting at 1 TR after stimulus onset are extracted, (ii) We apply successively the two training stages 'contrastive' and 'decode', and the inference step 'reconstruct' of the WAVE pipeline as provided by Wang et al. (2024b), with default parameters, with one exception: To accommodate the fact that NSD contains twice as many unique images per subject as the dataset used in Wang et al. (2024b), we have doubled the number of training steps for the 'contrastive' and 'decode' stages. This scaling factor was chosen following the approach of Wang et al. (2024b) on scaling training from 1 to 4 subjects simultaneously. We report metrics for the reconstructions obtained using the code [§] made available by Wang et al. (2024b). Figure 3 shows some of the NSD reconstructions obtained using this approach.

# B    TRAINING HYPERPARAMETERS

We train the brain module described in Section 2.1 and LoRA adapters of the pretrained latent diffusion model Xu et al. (2023) with AdamW optimizer and a maximum learning rate of $10^{-3}$, a weight decay of 0.01 and values of betas parameters of (0.9, 0.999). We append LoRA adapters to the diffusion model at all cross-attention layers, each with rank = 4 and alpha = 4. Besides, we apply a linear learning rate warmup during the first 1k training steps and then use a cosine decay schedule. Our model is trained for around 60k training steps using a total batch size of 320 on 8 A100 gpus. This typically results in a training time of 2.5 days. To optimize training efficiency, we use DeepSpeed ZeRO stage 2 Offload which offloads optimizer states and gradients to CPU and we train in float16 precision.

# C    ADDITIONAL QUANTITATIVE METRICS

In Table 6 we report single-trial quantitative metrics for all the four subjects, *i.e.*, without averaging the fMRI responses of the three image repetitions. We also report in the last five rows the average metrics across the four subjects.

To complete our experimental study, we trained Dynadiff on the beta values used in all previous image-decoding works on NSD. These values lack a time dimension thus we treat them as time series with a single time sample. Then, we adjust our brain module's architecture to take such data as input,

---

[§] https://github.com/ppwangyc/wave

Figure 7: Examples of images generated with Dynadiff, choosing among the best reconstructions.

by removing the timestep and temporal-aggregation layers (see Figure 2). We report our results in Table 4, where performances of other methods are sourced from their respective papers. While not specifically designed for beta values, we observe that our model surpasses DREAM (Xia et al., 2024a), UMBRAE (Xia et al., 2024b), MindBridge (Wang et al., 2024a) and MindEye1 (Scotti et al.,

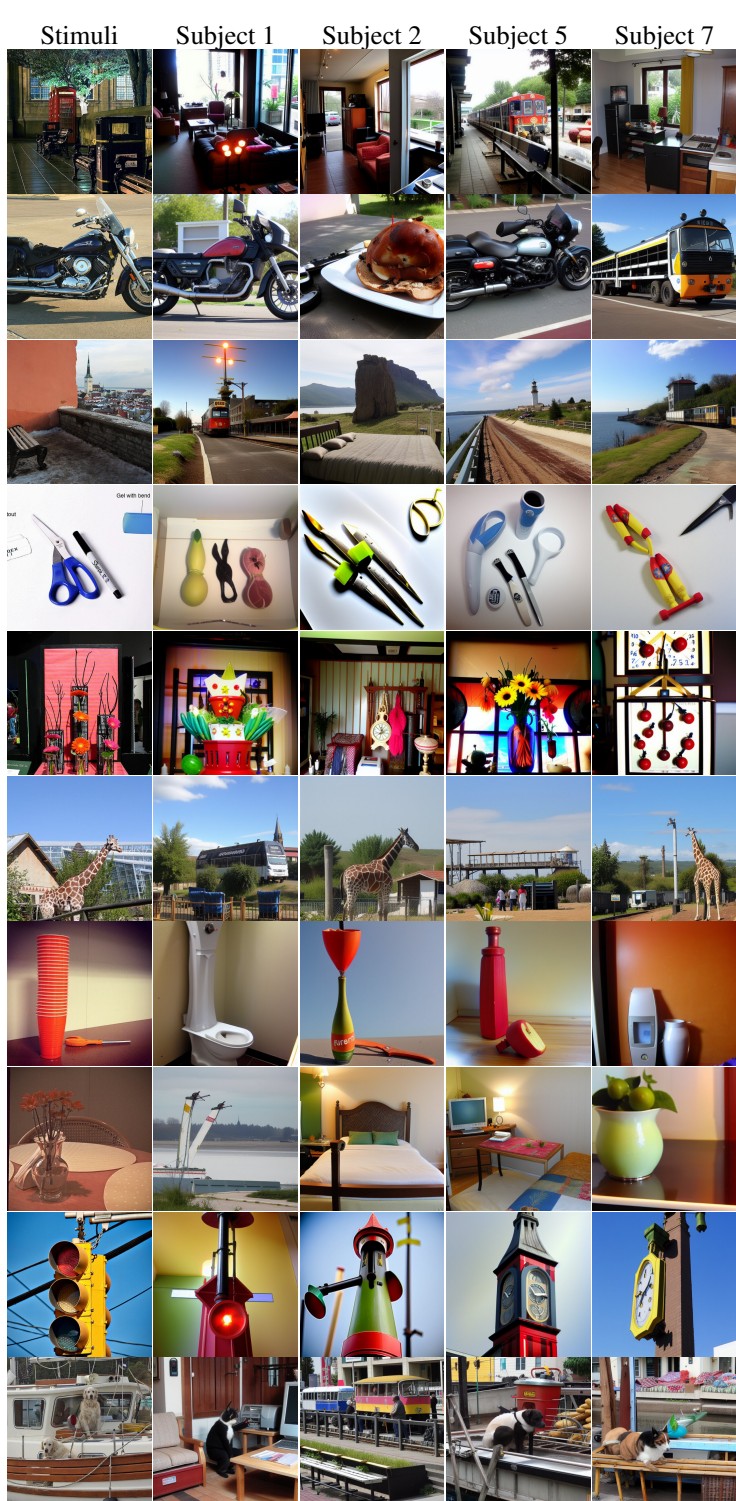

Figure 8: Examples of failure cases from Dynadiff.

2023). Furthermore, it performs on par with or only slightly below Psychometry, Neuropictor (Huo et al., 2024) and MindEye2 (Scotti et al., 2024), which all require more than one stage of training (unlike Dynadiff).

Table 4: Comparison of Dynadiff and baselines on averaged betas values from NSD (1.8 mm resolution, restricted to 'nsdgeneral'). Baseline results are reported directly from their respective introductory papers. As commonly done in prior studies, we report the average performance across all four subjects (1, 2, 5, and 7).

| Baseline | Low-level | | | | Semantic and High-level | | | |
|---|---|---|---|---|---|---|---|---|
| | ↑SSIM | ↑PixCorr | ↑AlexNet(2) | ↑AlexNet(5) | ↑CLIP-12 | ↑Incep | ↓Eff | ↓SwAV |
| DREAM (Xia et al., 2024a) | 0.33 | 0.27 | 93.90 | 96.70 | 94.10 | 93.40 | 0.64 | 0.42 |
| UMBRAE (Xia et al., 2024b) | 0.33 | 0.27 | 93.90 | 96.70 | 94.10 | 93.40 | 0.64 | 0.37 |
| MindBridge (Wang et al., 2024a) | 0.26 | 0.15 | 86.90 | 95.30 | 94.30 | 92.20 | 0.71 | 0.41 |
| MindEye1 (Scotti et al., 2023) | 0.32 | 0.31 | 94.67 | 97.80 | 94.05 | 93.75 | 0.65 | 0.37 |
| Psychometry (Quan et al., 2024) | 0.34 | 0.30 | 96.40 | 98.60 | **96.80** | **95.80** | 0.63 | 0.35 |
| NeuroPictor (Huo et al., 2024) | 0.38 | 0.23 | **96.55** | 98.38 | 93.35 | 94.50 | 0.64 | 0.35 |
| MindEye2 (Scotti et al., 2024) | **0.43** | **0.32** | 96.10 | **98.61** | 92.97 | 95.41 | 0.62 | **0.34** |
| Dynadiff | 0.37 | 0.21 | 95.72 | 98.11 | 94.09 | 95.03 | **0.61** | **0.34** |

Table 5: Preprocessing ablation

| Preprocessing type | Low-level | | | | Semantic and High-level | | | | | |
|---|---|---|---|---|---|---|---|---|---|---|
| | ↑SSIM | ↑PixCorr | ↑AlexNet(2) | ↑AlexNet(5) | ↑CLIP-12 | ↑Incep | ↓Eff | ↓SwAV | ↑mIoU | ↑DreamSim |
| Dynadiff w/ fMRIprep | **0.30** | 0.19 | 92.62 | 97.08 | 91.84 | 90.67 | 0.71 | 0.38 | 6.45 | 54.29 |
| Dynadiff w/ NSD prep. | **0.30** | 0.21 | **94.77** | **97.34** | **93.54** | **91.85** | **0.69** | **0.36** | **8.39** | **52.80** |

# D ADDITIONAL ABLATION

**Preprocessing types.** To measure the impact of data preprocessing on our model's performance, we provide an additional fMRI preprocessing ablation in Table 5, where we use `fMRIPrep` instead of NSD author's custom preprocessing. More precisely, we use `fMRIPrep` 23.2.0 with default settings to convert raw fMRI NSD data into MNI152NLin2009aAsym space. Then, brain volumes are mapped onto 'fsaverage5'. This process yields a time series of brain volumes for each recording run. Next, we eliminate low-frequency noise through a detrending step: a cosine-drift linear model is fitted to each voxel and subtracted from the raw signal. Each time series is then z-scored. Finally, the data is segmented into epochs starting 3s after stimulus onset and lasting for a total of 8s. As observed in Table 5, the NSD author's custom preprocessing enables superior image-decoding performance compared to using `fMRIprep` and projecting to 'fsaverage5'.

# E CROSS-SUBJECT DECODING WITH DYNADIFF

We evaluate the cross-subject decoding capabilities of Dynadiff in two ways.

First, we train Dynadiff jointly on the four subjects 1,2,5,7 of NSD. The goal is to determine whether a multi-subject model can perform competitively with single-subject models, while sharing most of the brain module's parameters across subjects. Indeed, instead of having four different independent brain

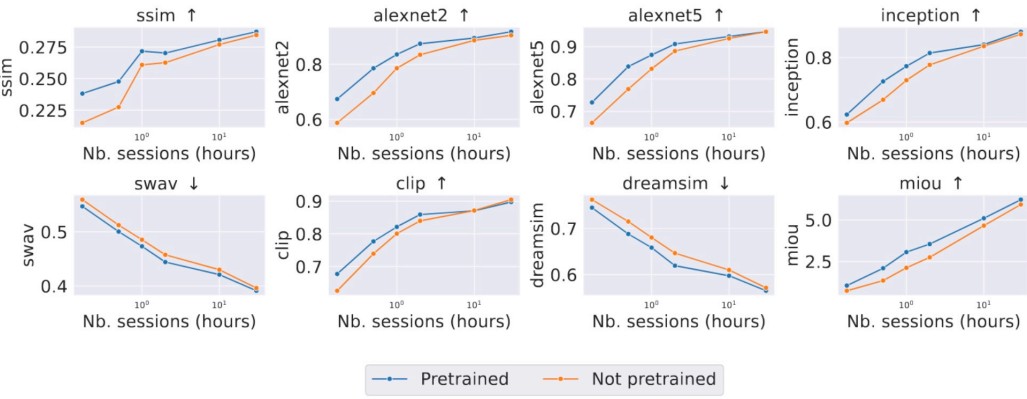

Figure 9: Cross-subject performance with varying amounts of data per subject.

Table 6: Comparison of Dynadiff and baselines for single-trial BOLD fMRI time series from NSD.

| Subjects | Models | Low-level | | | | Semantic and High-level | | | | | |
|---|---|---|---|---|---|---|---|---|---|---|---|
| | | ↑SSIM | ↑PixCorr | ↑AlexNet(2) | ↑AlexNet(5) | ↑CLIP-12 | ↑Incep | ↓Eff | ↓SwAV | ↑mIoU | ↓DreamSim |
| Subject 1 | Dynadiff | 0.35 | 0.25 | **97.30** | 98.84 | **93.02** | **91.00** | **0.69** | **0.36** | **8.44** | **52.58** |
| | WAVE | 0.16 | 0.08 | 70.37 | 77.46 | 76.65 | 72.59 | 0.85 | 0.55 | 2.18 | 69.40 |
| | MindEye1 | 0.32 | **0.32** | 94.18 | 96.57 | 91.49 | 89.33 | 0.72 | 0.40 | 7.94 | 57.34 |
| | MindEye2 | **0.37** | 0.27 | 96.06 | **98.03** | 90.46 | 88.67 | 0.72 | 0.38 | 8.36 | 56.05 |
| Subject 2 | Dynadiff | 0.35 | 0.22 | **97.42** | 98.87 | **93.43** | **91.63** | **0.67** | **0.36** | **8.97** | **51.94** |
| | Wave | 0.15 | 0.07 | 69.71 | 77.84 | 77.13 | 74.95 | 0.84 | 0.53 | 2.46 | 68.03 |
| | MindEye1 | 0.31 | **0.29** | 92.89 | 96.28 | 90.72 | 89.01 | 0.72 | 0.40 | 7.93 | 56.98 |
| | MindEye2 | **0.36** | 0.25 | 96.03 | 98.07 | 90.74 | 91.15 | 0.70 | 0.38 | 8.95 | 55.30 |
| Subject 5 | Dynadiff | 0.34 | 0.19 | **95.12** | 98.23 | **95.67** | **93.36** | **0.65** | **0.34** | **9.39** | **50.08** |
| | Wave | 0.15 | 0.08 | 69.28 | 79.50 | 77.75 | 74.01 | 0.84 | 0.52 | 2.51 | 67.98 |
| | MindEye1 | 0.31 | **0.25** | 90.25 | 95.54 | 92.79 | 90.92 | 0.72 | 0.40 | 8.00 | 56.47 |
| | MindEye2 | **0.36** | 0.22 | 93.16 | 97.67 | 92.42 | 91.11 | 0.69 | 0.37 | 8.63 | 54.35 |
| Subject 7 | Dynadiff | 0.34 | 0.19 | **93.44** | 96.84 | **92.00** | **89.21** | **0.71** | **0.38** | **7.22** | **55.47** |
| | Wave | 0.15 | 0.06 | 66.62 | 74.95 | 75.50 | 71.42 | 0.86 | 0.54 | 1.80 | 69.26 |
| | MindEye1 | 0.30 | **0.24** | 88.51 | 93.39 | 89.42 | 85.84 | 0.75 | 0.42 | 6.31 | 59.95 |
| | MindEye2 | **0.35** | 0.21 | 91.36 | 95.61 | 87.91 | 86.95 | 0.75 | 0.41 | 6.65 | 59.42 |
| Average | Dynadiff | 0.34 ± 0.00 | 0.21 ± 0.01 | **95.82 ± 0.82** | **98.20 ± 0.41** | **93.53 ± 0.67** | **91.30 ± 0.74** | **0.68 ± 0.01** | **0.36 ± 0.01** | **8.50 ± 0.41** | **52.52 ± 0.97** |
| | Wave | 0.15 ± 0.00 | 0.07 ± 0.00 | 68.99 ± 0.82 | 77.44 ± 0.94 | 76.76 ± 0.48 | 73.24 ± 0.78 | 0.85 ± 0.00 | 0.53 ± 0.01 | 2.24 ± 0.16 | 68.67 ± 0.38 |
| | MindEye1 | 0.31 ± 0.01 | **0.27 ± 0.01** | 91.45 ± 2.21 | 95.45 ± 0.62 | 91.11 ± 0.61 | 88.78 ± 0.92 | 0.73 ± 0.00 | 0.40 ± 0.00 | 7.55 ± 0.35 | 57.68 ± 0.67 |
| | MindEye2 | **0.36 ± 0.00** | 0.24 ± 0.01 | 94.15 ± 0.99 | 97.34 ± 0.50 | 90.38 ± 0.80 | 89.47 ± 0.89 | 0.71 ± 0.01 | 0.38 ± 0.01 | 8.15 ± 0.45 | 56.28 ± 0.95 |

modules, a multi-subject model only has a subject-layer and a timestep-layer per subject while the remaining layers are shared, thereby improving computational efficiency. The results are displayed in Table 8 and show that our multi-subject model performs on par with single-subject models.

Second, we investigate our model's generalization to unseen participants. For this, we pretrained Dynadiff on three of the four NSD subjects (2,5,7 and finetuned it on subject 1, varying the amount of data used for finetuning. To finetune the model on a new subject, we train subject- and timestep-layers from scratch, while other brain-module weights are finetuned. Figure 9 shows how performance increases with the number training sessions for: i) a single-subject model trained from scratch for subject 1 ('Not pretrained' setting), ii) the multi-subject model trained on 2,5 and 7 and fine-tuned on subject 1's data ('Pretrained' setting).

Notably, we see a performance gain when finetuning the pretrained model on a limited number of sessions, confirming that multi-subject pretraining reduces per-subject data requirements.

These experiments are in line with several previous works that tackled multi-subject training (Wang et al., 2024a; Xia et al., 2024b; Quan et al., 2024; Huo et al., 2024). These methods proposed different types of architectures and training recipes to account for the variability of brain activity between subjects. For example, MindBridge (Wang et al., 2024a) learns subject-invariant embeddings through a cycle consistency loss. UMBRAE (Xia et al., 2024b) uses a transformer with subject-specific tokens. Psychometry (Quan et al., 2024) further advances this with an Omni Mixture-of-Experts architecture. Neuropictor (Huo et al., 2024) trains a unified latent fMRI encoder across subjects, and follows with multi-subject pretraining. Unlike Dynadiff, these methods need to additionally finetune their pretrained multi-subject model on a given subject data to reach optimal performance for this subject.

Finally, MindEye2 (Scotti et al., 2024) shows good transfer capability to unseen subjects learning from a limited number of sessions and using averaged beta values. We demonstrate that the same phenomenon occurs with single-trial BOLD fMRI data in Figure 9.

# F VISUALIZATIONS

In Figures 7 and 8, we show additional reconstructions of Dynadiff for each subject. Figure 7 shows some of the best image reconstructions obtained, while Figure 8 displays failure cases. We observe that image stimuli with uncommon content in the training data tend to be reconstructed with reduced accuracy. For instance, the stimulus in the fourth row (a picture with scissors and pens) is rather unusual in the NSD dataset and is poorly reconstructed by Dynadiff. Besides, complex images containing many objects are challenging for Dynadiff, as shown by the last row of Figure 8.

Table 7: Comparison of Dynadiff and baselines for BOLD fMRI time series from NSD, averaged across same-image repetitions.

| Subject | Model | Low-level | | | | Semantic and High-level | | | | | |
|---|---|---|---|---|---|---|---|---|---|---|---|
| | | ↑SSIM | ↑PixCorr | ↑AlexNet(2) | ↑AlexNet(5) | ↑CLIP-12 | ↑Incep | ↓Eff | ↓SwAV | ↑mIoU | ↓DreamSim |
| Subject 1 | Dynadiff | 0.36 | 0.28 | **97.99** | **99.03** | 95.74 | **93.97** | **0.63** | **0.32** | 10.85 | **47.92** |
| | Wave | 0.20 | 0.09 | 75.79 | 86.19 | 84.44 | 81.91 | 0.79 | 0.50 | 3.84 | 64.05 |
| | MindEye1 | 0.31 | **0.37** | 95.93 | 97.45 | 93.69 | 92.23 | 0.69 | 0.37 | 8.98 | 54.04 |
| | MindEye2 | **0.37** | 0.29 | 97.44 | 98.92 | 92.91 | 92.83 | 0.67 | 0.35 | 10.32 | 52.59 |
| Subject 2 | Dynadiff | 0.36 | 0.25 | **97.78** | **98.96** | 95.23 | **94.50** | **0.63** | **0.33** | **10.96** | 48.28 |
| | Wave | 0.21 | 0.09 | 76.90 | 86.44 | 84.78 | 83.71 | 0.78 | 0.50 | 3.78 | 63.19 |
| | MindEye1 | 0.31 | **0.33** | 94.66 | 97.30 | 93.61 | 90.76 | 0.69 | 0.37 | 8.52 | 54.31 |
| | MindEye2 | **0.37** | 0.27 | 97.16 | 98.81 | 92.57 | 92.46 | 0.68 | 0.36 | 10.18 | 53.08 |
| Subject 5 | Dynadiff | 0.35 | 0.23 | **96.55** | **98.88** | 96.71 | 95.16 | **0.60** | **0.31** | **10.97** | **46.03** |
| | Wave | 0.22 | 0.09 | 76.94 | 86.34 | 86.06 | 83.22 | 0.78 | 0.48 | 3.93 | 62.26 |
| | MindEye1 | 0.30 | **0.29** | 92.68 | 96.52 | 94.13 | 91.84 | 0.68 | 0.37 | 8.92 | 53.53 |
| | MindEye2 | **0.37** | 0.25 | 95.81 | 98.68 | 94.10 | 93.06 | 0.65 | 0.34 | 9.73 | 51.42 |
| Subject 7 | Dynadiff | 0.35 | 0.22 | **95.64** | **98.25** | 95.11 | **93.34** | 0.64 | **0.34** | 9.49 | **49.59** |
| | Wave | 0.20 | 0.07 | 74.23 | 82.21 | 81.50 | 78.87 | 0.80 | 0.51 | 3.28 | 64.83 |
| | MindEye1 | 0.30 | **0.28** | 92.13 | 96.12 | 93.12 | 90.55 | 0.70 | 0.39 | 7.67 | 55.42 |
| | MindEye2 | **0.36** | 0.25 | 95.44 | 98.01 | 92.09 | 91.15 | 0.69 | 0.36 | **9.80** | 53.84 |
| Average | Our model | 0.35 ± 0.00 | 0.25 ± 0.01 | **96.99 ± 0.48** | **98.78 ± 0.16** | **95.70 ± 0.32** | **94.24 ± 0.34** | **0.63 ± 0.01** | **0.33 ± 0.01** | **10.57 ± 0.31** | **47.95 ± 0.64** |
| | Wave | 0.21 ± 0.00 | 0.08 ± 0.00 | 75.97 ± 0.64 | 85.29 ± 1.03 | 84.2 ± 0.96 | 81.93 ± 1.09 | 0.78 ± 0.00 | 0.5 ± 0.01 | 3.71 ± 0.15 | 63.58 ± 0.55 |
| | MindEye1 | 0.31 ± 0.00 | **0.32 ± 0.02** | 93.85 ± 0.76 | 96.85 ± 0.28 | 93.64 ± 0.18 | 91.34 ± 0.35 | 0.69 ± 0.00 | 0.38 ± 0.00 | 8.52 ± 0.25 | 54.33 ± 0.35 |
| | MindEye2 | **0.37 ± 0.00** | 0.26 ± 0.01 | 96.46 ± 0.43 | 98.60 ± 0.17 | 92.92 ± 0.37 | 92.37 ± 0.37 | 0.67 ± 0.00 | 0.35 ± 0.00 | 10.01 ± 0.12 | 52.73 ± 0.44 |

Table 8: Performance of Dynadiff when trained on multiple subjects.

| Multisubject model | Low-level | | | | Semantic and High-level | | | | | |
|---|---|---|---|---|---|---|---|---|---|---|
| | ↑SSIM | ↑PixCorr | ↑AlexNet(2) | ↑AlexNet(5) | ↑CLIP-12 | ↑Incep | ↓Eff | ↓SwAV | ↑mIoU | ↑DreamSim |
| Subject 1 | 0.34 | 0.17 | 96.29 | 98.57 | 92.66 | 90.73 | 0.70 | 0.37 | 8.10 | 52.87 |
| Subject 2 | 0.35 | 0.19 | 96.24 | 98.68 | 93.46 | 92.06 | 0.69 | 0.37 | 8.45 | 52.34 |
| Subject 5 | 0.35 | 0.17 | 94.72 | 98.64 | 95.49 | 92.79 | 0.66 | 0.36 | | 51.02 |
| Subject 7 | 0.33 | 0.13 | 92.61 | 96.42 | 91.31 | 89.15 | 0.73 | 0.38 | 7.14 | 55.84 |
| Average | 0.34 ± 0.01 | 0.17 ± 0.01 | 94.96 ± 0.87 | 98.08 ± 0.55 | 93.23 ± 0.87 | 91.18 ± 0.8 | 0.69 ± 0.01 | 0.37 ± 0.0 | 8.17 ± 0.39 | 53.02 ± 1.02 |

# G  LLM USAGE

We used LLMs to polish writing and improve the flow of sentences. We did not use LLMs for any other purpose.

