# OpenReview forum: "Dynadiff: Single-stage Decoding of Images from Continuously Evolving fMRI"
_ICLR.cc/2026/Conference — Submitted to ICLR 2026_

### Official Review · Reviewer_jcgu · 2025-10-30

**Soundness:** 2
**Presentation:** 3
**Contribution:** 2
**Rating:** 4
**Confidence:** 2

**Summary:**

This paper introduces Dynadiff, a single‑stage diffusion‑based decoder that conditions a frozen latent diffusion model on continuous BOLD fMRI time series via a learned “brain module,” thereby avoiding beta averaging and multi‑stage pipelines common in prior work. Experiments are conducted on NSD without averaging repetitions at train/test time. The method improves especially on high‑level/semantic metrics for single‑trial time series, and analyzes temporal generalization by shifting decoding windows to probe when reconstructions best align with the viewed image. The paper frames these results as a step toward real‑time decoding, though all data are static‑image sequences from NSD rather than natural movies.

**Strengths:**

- Dynadiff jointly trains only two components (i) a brain module (MLP) that maps brain activity to the diffusion model’s conditioning tokens and (ii) LoRA adapters on cross‑attention while freezing the rest of the generator. No pretrained fMRI encoder, no alignment stage to a fixed embedding space, and no post‑generation selection/refinement are required. This reduces engineering complexity.

- Training and evaluation use single‑trial BOLD windows from the standard‑resolution NSD time series; repetitions are not averaged, and preprocessing deliberately keeps temporal information. This directly addresses “time‑collapse” concerns in prior NSD decoders.

- Competitive to SOTA on single‑trial time series, especially semantic metrics.

- The temporal generalization was evaluated by shifting the input window during testing. The results show that a model trained at a fixed relative time generalizes best near that time, whereas models specialized for each δ-offset can still decode the current image at later time points.These findings are consistent with the latency characteristics of the hemodynamic response function.

- Ablations on window length, brain‑module design, and which generator layers to adapt support design choices. Hyperparameters and training compute are specified.

**Weaknesses:**

- Gains vs. MindEye2 are uneven and sometimes within SEM; low‑level metrics drop. In Tab. 1, SSIM and PixCorr are below MindEye2. Improvements on AlexNet(2/5), Inception, SwAV, mIoU are reported, but several margins are modest and some overlap with SEM. It is thus difficult to determine whether these differences are genuinely meaningful. In addition, when all methods are evaluated on averaged beta values (the dominant NSD setting), Dynadiff is competitive but trails on several metrics.

- The caption of Fig. 4 says “Real‑time decoding,” but all experiments use static‑image sequences from NSD; the time‑resolved results come from shifted windows. There is no dynamic stimuli such as natural‑movie dataset evaluation. Also, rationale for not evaluating on videos is unconvincing. The Discussion argues that prior video‑decoders often rely on time‑collapsed betas, multi‑stage training, and smaller data; however, similar caveats also apply to NSD baselines and are not specific reasons to avoid at least a small‑scale video test. Given the paper’s emphasis on time‑resolved decoding, a movie benchmark (and comparing their method with the methods for previous movie reconstruction methods) seems warranted.

- The brain module has ~400M parameters; inference uses DDIM 20 steps. While training details are thorough, the paper does not report per‑trial inference time or throughput relative to NSD’s TR=1.3 s, leaving the “real‑time” feasibility unclear.

**Questions:**

- Clarify “real‑time.” What is the wall‑clock latency from receiving a new TR to outputting an image with 20 DDIM steps? Can the model decode per‑TR at TR=1.3 s on a single GPU?
- Metric arrows for DreamSim (↑/↓) are not fully consistent across tables (see Tables 1 and 5).
- Table 8 appears to have a column misalignment for Subject 5 (mIoU missing).

---

### Official Review · Reviewer_v9Ga · 2025-11-01

**Soundness:** 2
**Presentation:** 3
**Contribution:** 2
**Rating:** 4
**Confidence:** 4

**Summary:**

The paper introduces Dynadiff, a single-stage diffusion model for reconstructing images directly from time-resolved fMRI signals. Unlike previous multi-stage approaches, Dynadiff jointly trains a lightweight brain-to-diffusion conditioning module with LoRA-adapted diffusion layers. The method avoids time-collapsing preprocessing, supports temporal decoding, and achieves state-of-the-art results on the NSD dataset for both semantic and perceptual metrics. Ablations show the importance of temporal modeling and LoRA-based fine-tuning.

**Strengths:**

* The single-stage training pipeline is a clear improvement over existing multi-stage frameworks.
* Demonstrates robust time-resolved reconstruction from continuous BOLD signals.
* Includes ablations on time-window duration, brain module design, and diffusion tuning strategies.
* Well-written and clearly motivated, especially regarding the challenges of time-collapsed preprocessing.

**Weaknesses:**

* While the single-stage design is elegant, the core idea, jointly fine-tuning an fMRI encoder with a diffusion model, remains conceptually close to prior fMRI-to-image diffusion frameworks.
* Table 1 employs a customized fMRI preprocessing pipeline while comparing against baselines trained on time-collapsed data, making the reported performance gains difficult to interpret. Moreover, the comparison omits recent time-resolved decoders such as Neuropictor (Huo et al., 2024), which weakens the fairness of the evaluation.
* In Table 4, where multiple strong baselines are evaluated under the same setting, Dynadiff shows no clear or consistent advantage, casting doubt on its claimed state-of-the-art performance.
* The evaluation is limited to four NSD subjects, without testing cross-subject or cross-dataset generalization, leaving its robustness and scalability uncertain.

**Questions:**

* How fair is the comparison in Table 1, given that other baselines were not trained under the same preprocessing pipeline?
* How sensitive are the results to preprocessing choices such as ROI selection, and normalization?
* Why are time-resolved baselines (e.g., Neuropictor) excluded from the main comparison?
* Can Dynadiff generalize to new participants or datasets without subject-specific retraining?
* What factors explain the lack of improvement in Table 4 despite the proposed architectural changes?
* How might this framework be extended or validated for continuous video decoding rather than static images?

---

### Official Review · Reviewer_hpZN · 2025-11-01

**Soundness:** 2
**Presentation:** 3
**Contribution:** 2
**Rating:** 4
**Confidence:** 4

**Summary:**

The paper proposes **Dynadiff**, a “single-stage” fMRI-to-image system that conditions a pretrained diffusion model directly on short fMRI time windows, adding LoRA adapters to cross-attention. The authors argue (i) prior work “collapses time” by using GLM betas, (ii) their time-resolved approach yields better temporal fidelity, and (iii) a one-stage objective simplifies training. Experiments are on NSD with multiple metrics and temporally shifted windows.

**Strengths:**

# Strengths

* **Interesting temporal visualization**: The time-shift analyses (e.g., Fig. 4) are engaging and make the dynamics tangible.
* **Simplified training story**: Collapsing multi-stage pipelines into a single training objective is an appealing engineering direction.
* **Reasonably broad metrics**: CLIP/feature metrics, segmentation mIoU, and qualitative examples provide multiple views of performance.

**Weaknesses:**

# Weaknesses & Detailed Comments

## A. Methodology/Claims

1. **“Time-series modeling” is shallow relative to the claim.**
   The core temporal handling appears to be per-timestep linear transforms plus a **single temporal aggregation layer**. This is not a genuine temporal model (no temporal attention, SSM/RNN, or FIR deconvolution) and does not convincingly support the claim that prior work “completely discards the time dimension.” Please either temper the claim or compare against lightweight temporal baselines (1D convs, attention, S4/SSM) to show real sequence modeling helps.

2. **Over-sized mapper vs. data; missing capacity control.**
   The brain-to-token mapper is very large given per-subject data. There’s little regularization beyond dropout, and no capacity ablations (e.g., lowering hidden dims, low-rank constraints, weight sharing across timesteps). This raises overfitting and stability concerns and limits interpretability.

3. **Preprocessing rationale is thin.**
   Avoiding nuisance regression/high-pass to “avoid assumptions” is not sufficient; it may let the model pick up motion/physio artifacts. The later “preprocessing ablation” changes multiple factors at once (pipeline and surface space), making attribution ambiguous.

## B. Experimental Design & Evaluation

1. **Potential leakage from hemodynamic overlap in the *main table*.** (important)
   NSD uses rapid event-related timing (SOA ~4 s). With multi-second windows centered near the HRF peak, test windows inevitably contain BOLD from neighboring stimuli—some likely from the training split if images are interleaved. Your time-resolved figures correctly use a *run-wise* split, but the headline table seems to rely on the standard interleaved split, risking contamination that favors time-series methods. Maybe consider recompute the main table under a run-wise (or otherwise no-overlap) split, or decode from windows that minimize neighbor contamination; add FIR/GLM-deconvolved controls.

2. **Baselines are not fully matched to your setting.**
   WAVE appears evaluated on a different start/duration window and possibly different ROIs; MindEye baselines flattened time without temporal modeling. These choices may systematically depress them.

3. **Time-series vs. betas: claim not supported.** (important)
   When you switch from betas to raw time-series, high-level metrics generally **drop**, and Dynadiff does not surpass MindEye2 on betas. The supposed “time-collapsing problem” is asserted but not shown to harm decoding.
Maybe you can demonstrate a concrete failure mode on betas (e.g., a class of stimuli where betas underperform), or show that adding temporal modeling on *deconvolved* timecourses surpasses betas under matched evaluation.


## C. Neuroscience Interpretation

1. **“Dynamic coding” interpretation is confounded by the HRF.**
    The fact that windows shifted to “previous/next image” decode those images can be explained by HRF overlap in fast event-related designs; it does not by itself evidence a novel neural dynamic. maybe you could consider (i) cross-time generalization matrices (train at δ, test across δ′) with **deconvolved** signals; (ii) a no-overlap control (preceding/following stimuli from held-out runs or categories); (iii) show pre-onset decoding at chance.

## D. Relation to Prior Work

 **Positioning vs. recent temporal/video decoders is incomplete.**
    The claim that prior SOTA “completely discards time” is too strong. Recent works also model temporal structure across subjects/videos, please consider comparing with them

**Questions:**

# Questions for Authors

* How sensitive is performance to the mapper’s capacity and to the window start/duration?
* Did you try temporal attention or FIR deconvolution before aggregation?

# Suggestions
* Leakage control: Recompute headline results under no-overlap splits and/or deconvolved signals.
* Matched baselines: Align windows/ROIs; add a time-aware MindEye baseline.
* Temper or justify claims: Maybe rephrase the “completely discards time” statement and substantiate the “one-stage” benefits.

---

### Official Review · Reviewer_LP21 · 2025-11-02

**Soundness:** 2
**Presentation:** 3
**Contribution:** 2
**Rating:** 4
**Confidence:** 3

**Summary:**

This paper introduces Dynadiff, a novel brain-to-image decoding model designed to reconstruct visual stimuli from fMRI data. The authors identify two primary limitations in current approaches: (1) the reliance on complex, multi-stage training pipelines , and (2) the common practice of "time-collapsing" fMRI data into static 'beta values', which discards temporal information。 Dynadiff addresses both issues by proposing a single-stage diffusion model that works directly on continuously evolving fMRI time-series.  The authors demonstrate experimentally on the Natural Scenes Dataset (NSD) that Dynadiff outperforms state-of-the-art models (like MindEye2 and WAVE) on the task of time-resolved, single-trial fMRI decoding.

**Strengths:**

1. The proposed method is simple and straightforward. Experimental results shows that compared to state of the art model the proposed method achieved comparable performance using only one stage training.
2. The experiment analysis is completed and interesting, which could provide useful insight to the community.
3. The experimental evaluation is thorough and robust.

**Weaknesses:**

1. Architectural novelty. While the pipeline is novel in its simplicity, the components are standard. The brain module is essentially a large MLP , and the finetuning method is LoRA. This is not a major flaw, as the contribution lies in the effective composition and problem formulation, but the architectural novelty itself is moderate.
2. Fairness of Time-Series Baselines. Models like MindEye1 and MindEye2 were explicitly designed for static beta values. The authors state they adapted these models by "flatten this window into a vector of size $C \times T$"22. This is a very naive way to incorporate temporal data and is likely a highly suboptimal adaptation that puts the baselines at a significant disadvantage. A simple flattening operation discards all explicit temporal structure. A fairer comparison might involve adapting the baselines with a more sophisticated temporal adapting method before feeding the data to their respective mapping networks.

* In the line 83 I believe the Seeing beyond the brain paper does not use contrastive learning.

**Questions:**

1. The brain module is very large (~400M parameters). Your ablation in Table 2 justifies the types of layers used, but not their size. Did you experiment with smaller brain modules? How much does performance degrade if the brain module is, for example, 100M or 50M parameters? Is this large size truly necessary?

2. Could you please elaborate on the adaptation of the MindEye baselines? Do you agree that simple flattening  is a weak baseline for temporal data? How confident are you that Dynadiff's superior performance in Table 1 is due to its architectural design rather than this suboptimal baseline adaptation?

---

### Meta-Review · Area_Chair_TWcy · 2025-12-17

**Summary:**

This paper introduces Dynadiff, a single-stage diffusion model for reconstructing images from time-resolved fMRI signals. The work addresses two stated limitations of prior approaches: (1) reliance on complex multi-stage training pipelines, and (2) "time-collapsing" of fMRI data that discards temporal information. The authors demonstrate results on the Natural Scenes Dataset (NSD) showing competitive or superior performance on high-level semantic metrics for single-trial fMRI decoding.
All four reviewers assigned a score of 4 (marginally below acceptance threshold), indicating consistent concerns about the contribution despite recognizing some merits. The reviews converge on several critical issues:

1. Fair Comparison & Experimental Design:

* Multiple reviewers (hpZN, v9Ga, LP21) raise concerns about baseline comparisons. The main results (Table 1) compare Dynadiff using custom preprocessing against baselines trained on time-collapsed betas, making it difficult to attribute performance gains to architectural innovations versus preprocessing choices.
* Reviewer LP21 notes that adapting baselines (MindEye1/2) via "naive flattening" of temporal windows creates a "highly suboptimal" comparison that "puts baselines at significant disadvantage."
* Reviewer hpZN identifies a potential data leakage issue: With rapid event-related timing (SOA ~4s) and multi-second fMRI windows, test windows likely contain BOLD signal from neighboring training stimuli, potentially inflating time-series method performance.
* Reviewer v9Ga observes that in Table 4, where methods use the same preprocessing, Dynadiff shows "no clear or consistent advantage," undermining claims of state-of-the-art performance.

2. Limited Technical Novelty:

* Reviewers LP21, v9Ga, and hpZN note that while the simplified pipeline is appealing, the core components (large MLP brain module, LoRA fine-tuning) are standard.
* Reviewer hpZN criticizes the temporal modeling as "shallow" - using per-timestep linear transforms and single aggregation layer rather than genuine sequence modeling (no temporal attention, RNN, or state-space models).
* The conceptual contribution of jointly fine-tuning an fMRI encoder with a diffusion model remains "close to prior frameworks" (v9Ga).

3. Unsupported or Overclaimed Contributions:

* "Time-collapsing problem": Reviewers hpZN and v9Ga note this is asserted but not demonstrated. When switching from betas to time-series, high-level metrics often drop, and Dynadiff does not surpass MindEye2 on betas.
* "Real-time decoding": Reviewer jcgu points out this claim is unsupported - all experiments use static images from NSD, no actual video/dynamic stimuli evaluation, and no reported inference latency relative to TR=1.3s.
* Dynamic neural coding interpretation: Reviewer hpZN notes this is confounded by hemodynamic response overlap in the rapid design and requires deconvolution controls.

4. Missing Comparisons & Limited Scope:

* Reviewers v9Ga and hpZN note absence of recent time-resolved decoders (e.g., Neuropictor, Huo et al. 2024).
* Reviewer jcgu emphasizes that given the paper's emphasis on temporal decoding, the lack of movie/video benchmarks is a significant limitation. The rationale provided is unconvincing since similar caveats apply to NSD baselines.
* Evaluation limited to 4 NSD subjects with no cross-subject or cross-dataset generalization (v9Ga).

5. Methodological Concerns:

* The brain module has ~400M parameters with limited regularization, raising overfitting concerns. No capacity ablations provided (hpZN, LP21).
* Preprocessing rationale is weak - avoiding nuisance regression may allow motion/physiological artifacts to influence results (hpZN).
* Performance gains are often modest and within standard error margins, with some low-level metrics (SSIM, PixCorr) actually degrading (jcgu).

**Reviewer Concerns:**

There was not response by the authors. Thus all issues remain unresolved.

**Reviewer Scores:**

Because there was no response, no change in scores is expected.

---

### Decision · Program_Chairs · 2026-01-26

Reject